

# Genome-wide analysis of the WRKY gene family in drumstick (*Moringa oleifera* Lam.)

Junjie Zhang[1,2,3,4,*], Endian Yang[4,*], Qian He[3,4], Mengfei Lin[1,2,3,4], Wei Zhou[1,2,3,4], Ruiqi Pian[1,2,3,4] and Xiaoyang Chen[1,2,3,4]

[1] State Key Laboratory for Conservation and Utilization of Subtropical Agro-bioresources (South China Agricultural University), Guangzhou, China
[2] Guangdong Key Laboratory for Innovative Development and Utilization of Forest Plant Germplasm, Guangzhou, China
[3] Guangdong Province Research Center of Woody Forage Engineering Technology, Guangzhou, China
[4] College of Forestry and Landscape Architecture, South China Agricultural University, Guangzhou, China
[*] These authors contributed equally to this work.

Corresponding author
Xiaoyang Chen, 769495781@qq.com, xychen@scau.edu.cn

## ABSTRACT

WRKY proteins belong to one of the largest families of transcription factors. They have important functions in plant growth and development, signal transduction and stress responses. However, little information is available regarding the WRKY family in drumstick (*Moringa oleifera* Lam.). In the present study, we identified 54 *MoWRKY* genes in this species using genomic data. On the basis of structural features of the proteins they encode, the *MoWRKY* genes were classified into three main groups, with the second group being further divided into five subgroups. Phylogenetic trees constructed from the sequences of WRKY domains and overall amino acid compositions derived from drumstick and Arabidopsis were similar; the results indicated that the WRKY domain was the main evolutionary unit of *WRKY* genes. Gene structure and conserved motif analysis showed that genes with similar structures and proteins with similar motif compositions were usually clustered in the same class. Selective pressure analysis indicated that although neutral evolution and positive selection have happened in several *MoWRKY* genes, most have evolved under strong purifying selection. Moreover, different subgroups had evolved at different rates. The levels of expression of *MoWRKY* genes in response to five different abiotic stresses (salt, heat, drought, $H_2O_2$, cold) were evaluated by reverse transcription polymerase chain reaction (RT-PCR) and quantitative RT-PCR (qRT-PCR), with the results indicating that these genes had different expression levels and that some may be involved in abiotic stress responses. Our results will provide a foundation for cloning genes with specific functions for use in further research and applications.

## INTRODUCTION

Transcription factors (TFs), which bind to specific DNA motifs, are important in regulating gene expression and controlling various important biological processes (*Smith & Matthews, 2016*). Out of numerous families of TFs, the WRKY gene family, named after a conserved

WRKY domain, is one of the largest, and it is known to be involved in a range of plant processes from germination to senescence (*Chen et al., 2012*; *Eulgem et al., 2000*; *Pandey & Somssich, 2009*; *Ulker & Somssich, 2004*). WRKY genes were first identified in plant species (*Ishiguro & Nakamura, 1994*) and originally thought to be plant-specific (*Eulgem et al., 2000*). However, in recent years WRKY proteins have been identified in non-plant species, such as *Giardia lamblia*, *Dictyostelium discoideum* and so on (*Li et al., 2016*; *Zhang & Wang, 2005*). The WRKY domain contains about 60 amino acid residues, comprising a highly conserved short amino acid sequence, WRKYGQK, at the N-terminus and an adjacent $C_2H_2$ or $C_2HC$ zinc finger structure (*Eulgem et al., 2000*). Depending on the number of WRKY domains and the type of zinc finger, the WRKY family can be divided into three main groups. Group I contains two WRKY domains and the $C_2H_2$ zinc finger type. Group II contains one WRKY domain and a $C_2H_2$ type zinc finger motif; this group can be further divided into five subgroups, IIa, IIb, IIc, IId and IIe. The WRKYs with a single WRKY domain and a $C_2HC$ zinc-finger structure belong to group III (*Eulgem et al., 2000*; *Goel et al., 2016*; *Li et al., 2017*).

In recent years, with the development of novel sequencing technologies and bioinformatics, genome-wide WRKY analysis has been performed in many plant species including *Populus trichocarpa* (*He et al., 2012*), *Pyrus bretschneideri* (*Huang et al., 2015*), *Citrus* (*Ayadi et al., 2016*), *Glycine max* (*Song et al., 2016*), *Daucus carota* (*Li et al., 2016*) and so on. Studies on WRKY identification and functional analysis have shown that WRKY TFs play significant roles in signaling and regulation of expression during various biotic and abiotic stresses. In banana, MaWRKY26 activated jasmonic acid biosynthesis and enhanced cold tolerance in the fruit (*Ye et al., 2016*). In wheat, *TaWRKY49* and *TaWRKY62* were shown to participate in the defense response against the fungal pathogen *Puccinia striiformis* f. sp tritici (Pst), *TaWRKY49* was shown to be a negative regulator and *TaWRKY62* a positive regulator of wheat's HTSP resistance to Pst (*Wang et al., 2017*). WRKY TFs have also been implicated in the modulation of plant development. In the poplar *Populus trichocarpa*, *PtrWRKY19* may function as a negative regulator of pith secondary wall formation (*Yang et al., 2016*). In foxtail millet, map-based cloning, combined with high-throughput sequencing, revealed that LP1, which encodes a novel WRKY TF, regulates panicle development (*Xiang et al., 2017*). WRKY TFs have also been shown to regulate the production of several secondary metabolites such as phenolic compounds including lignin, flavanols and tannins. In *Arabidopsis*, *AtWRKY23* regulates the production of flavanols in auxin inducible manner (*Grunewald et al., 2008*; *Grunewald et al., 2013*). In rice, *OsWRKY76* activates cold stress tolerance but suppresses PR genes and production of phytoalexins like terpene and the phenylpropanoid sakuranetin (*Yokotani et al., 2013*). In *Withania somnifera*, WsWRKY1 binds to W-box sequences in promoters encoding squalene synthase and squalene epoxidase, indicating that it has a direct role in the regulation of the triterpenoid pathway (*Singh et al., 2017*). What's more, the WRKYs always work interaction with other proteins, such as *PeWRKY83* could interact with *PeVQ* proteins in moso bamboo (*Wu et al., 2017*) and physical interaction of WRKY75 with DELLA repressors were also found in *Arabidopsis thaliana* (*Zhang, Chen & Yu, 2018*).

*Moringa oleifera* Lam., commonly known as drumstick, belongs to the monogeneric family *Moringaceae* (*Ramachandran, Peter & Gopalakrishnan, 1980*). This species is widely cultivated in tropical and sub-tropical areas and has a long history of traditional medicine and culinary uses (*Anwar et al., 2007*; *Zhang et al., 2017*). Drumstick is considered to be a fast-growing tree species, and also it is a drought tolerant plant that can be grown in diverse soils except those that are waterlogged; it may also become important for biofuel production and has been used in a variety of industrial applications (*Popoola & Obembe, 2013*; *Shih et al., 2011*). Studies on drumstick transcription factors have hitherto rarely been reported because of a lack of genomic data for this species. The publication of the drumstick genome draft database (*Tian et al., 2015*) provides resources with which to carry out bioinformatics-based identification and analysis of WRKY TFs. In the present study, we have used these genomic resources to identify members of the WRKY gene family in drumstick and correlated their expression with various stress responses. We carried out a detailed study of the drumstick WRKY gene family, including gene classification, phylogenetic analysis, determination of structural organization and conserved motif composition, and assessed the selective pressures that have acted on different members of this family.

## MATERIAL AND METHODS

### Sequence database searches

The complete genome and proteome sequences and General Feature Format (GFF) file for *Arabidopsis* were downloaded from TAIR (http://www.arabidopsis.org). The annotated drumstick genome sequences were provided by Yunnan Agricultural University. A WRKY-domain Hidden Markov Model (HMM) Profile, which was downloaded from Pfam (http://pfam.xfam.org/), was used as a query with which to search all of the annotated proteins in the drumstick genome with an $E$ value cut-off of 1E−5. The candidates selected using HMMER were examined to determine whether they had typical features of WRKY proteins by employing the Pfam database. Finally, the CD-HIT program and the Pfam database were used to eliminate duplicate and incomplete sequences. Non-overlapping WRKY protein sequences were used for further analysis.

### Multiple sequence alignment and phylogenetic analyses

The conserved WRKY domains of *MoWRKY* genes obtained using manual inspection in the Pfam program were aligned using ClustalX 1.83 software. Phylogenetic analysis including seven representative domains from *Arabidopsis* was carried out to obtain better classifications of the different clades by applying the Neighbor-Joining method with 1,000 bootstrap replicates using MEGA 6 software.

### Gene structure and motif composition analysis

Analysis of the exon-intron organization of *Mo* WRKYs was performed by comparing the coding sequences of *Mo* WRKYs with their corresponding genomic sequences using GSDS software (*Hu et al., 2015*). Conserved motifs in each WRKY protein were investigated using the Multiple Expectation Maximization for Motif Elucidation (MEME) online program:

http://meme-suite.org/. The following parameters were employed in analysis: maximum number of motifs 20; minimum motif width 6; maximum motif width 50.

## Promoter *cis-acting* elements analysis of *MoWRKYs*

The promoter sequences, 1.5kb upstream of the translation start site, of the *MoWRKY* genes were obtained from drumstick genome. PlantCARE (*Lescot et al., 2002*) was used to analyse the *MoWRKY* gene promoters and identify their cis-acting elements.

## Tests for selective pressure

The multiple sequence alignment of drumstick MoWRKY proteins was carried out using ClustalW with default parameters. Then the sequences were trimmed to reduce gap penality. DNAMAN was used to search for nucleotide sequences encoding additional WRKY proteins, with the aligned MoWRKY protein sequences as guides. The synonymous ($K_S$) and nonsynonymous ($K_a$) substitution rates were calculated with the YN00 program in PAML4.9 with default parameters (*Yang, 2007*).

## Expression analysis

To investigate the patterns of expression of *MoWRKY* genes under normal and abiotic stress conditions, seedlings of drumstick were cultivated in potting soil at 25 °C under 14: 10 h light: dark conditions in a growth chamber for 20 days before treatment. For salt and oxidative stress treatments, seedlings were sprayed for 12 h with, respectively, 150 mM NaCl and $H_2O_2$ solution. Cold and heat stress were applied by transferring plants to a climate chamber at, respectively, 4 °C and 42 °C for 12 h. Drought stress was induced by withholding water for 2 weeks. Each treatment consisted of three replicates. After stress treatments, total RNA was isolated from leaf, stem, stem tip and root tissues of each seedling using a Total RNA Kit (OMEGA, Guangzhou, China). Total RNA was reverse transcribed into cDNA using a PrimeScript RT Master Mix (Perfect) Real Time Kit (Takara, Dalian, China). Gene specific primers were designed using Primer 5.0 and the *RPL* gene was used as a reference (*Deng et al., 2016*). Expression of all *MoWRKY* genes was examined by RT-PCR and products from each sample were analyzed using a 1% agarose gel. Among all *MoWRKY* genes, nine genes belonging to different subgroups were selected for analysis of gene expression levels using qRT-PCR according to the method described in *Wei et al. (2016)* and *RPL* was amplified as a reference gene (*Deng et al., 2016*). Relative expression levels were evaluated using the $2^{-\Delta\Delta CT}$. Three technical replicates were conducted for test and reference genes of each sample to obtain precise and reproducible results. Statistical analysis was carried out using SPSS 19.0 software (SPSS Inc., Chicago, IL, USA), Duncan's multiple range test was used to detect differences among means. A *p*-value <0.05 was considered significant.

# RESULTS

## Identification of WRKY family members in the drumstick genome

To identify all the *WRKY* genes in the drumstick genome, we employed the HMM profile of the WRKY domain (PF03106) as a query to search against the drumstick genome

database using HMMER 3.0 and BLAST. A total of 54 nonredundant genes (Table 1) were identified as *WRKY* genes and a unique name was assigned to each drumstick *WRKY* gene, consisting of two italic letters denoting the source organism and sequential numbers: *MoWRKY1* to *MoWRKY54*. All the putative 54 *WRKY* genes were further analyzed to confirm the presence of the WRKY domain and all of them were annotated with gene ontology (GO) terms (File S1). Fifty-three *MoWRKY* genes containing complete WRKY domains were identified; only one gene (*MoWRKY50*) lacked a complete domain. The highly conserved domain WRKYGQK was present in 52 of the MoWRKY proteins, whereas the remaining one (MoWRKY24) contained a WRKYGKK domain. The lengths of the MoWRKY proteins ranged from 106 (*MoWRKY24*) to 834 (*MoWRKY3*) amino acids; the average length was 391 amino acids.

## Phylogenetic relationship and classification of *MoWRKY* genes

The most prominent structural feature of WRKY genes is a conserved WRKY domain; there is also a zinc-finger motif. Among the 54 MoWRKY proteins identified, nine MoWRKY proteins contained two WRKY domains; since one MoWRKY protein did not have a complete WRKY domain, a total of 62 WRKY domains were found in this study. In each protein that contained two WRKY domains, we designated these domains by the WRKY name plus N or C for the N-terminal or C-terminal domain respectively. In order to examine phylogenetic relationships and classify all 62 MoWRKY domains, a phylogenetic tree based on conserved WRKY domains was constructed. Representative WRKY domains from Arabidopsis were used in our analysis, and the candidate domains were obtained from *Diao et al. (2016)* and *Li et al. (2016)*. Figure 1 shows a multiple sequence alignment of the 62 WRKY domains. Three major groups were identified, as previously described in poplar (*He et al., 2012*), pepper (*Diao et al., 2016*) and carrot (*Li et al., 2016*). Additionally, several subgroups were apparent on the basis of the phylogenetic analysis.

Group I contained 10 WRKY proteins, of which all contain two WRKY domains except for MoWRKY10. This member might have lost the N-terminal WRKY domain during evolution, since its single WRKY domain showed high similarity to MoWRKY1C, which is located in the C-terminal WRKY domain clade, suggesting a common origin for these two domains. Group II had the largest numbers of WRKY proteins and was divided into five major subgroups: IIa, IIb, IIc, IId and IIe. Subgroup IIa (three members) and IIb (eight members) were two subgroups in the same branch, while subgroup IId (five members) and IIe (seven members) were derived from one clade. Subgroup IIc, with 14 members, was more similar to group I than to any other subgroups according to the phylogenetic analysis. Furthermore, six WRKY domains belonged to group III, which is widely considered to be the most advanced in terms of evolution and the most relevant to adaptability (*Dou et al., 2016*; *Kalde et al., 2003*; *Huang et al., 2016*). Comparing the two phylogenetic trees, constructed for MoWRKY domains and genes, similar groups and subgroups were identified, though the classifications of a few members were different (Fig. 1 and Fig. S1), indicating that the conserved WRKY domain is an important unit in WRKY proteins.

**Table 1 Full-length WRKY genes identified from drumstick genome.**

| Class | Gene name | Annotation ID | Conserved motify | Zinc finger |
|---|---|---|---|---|
| I | MoWRKY43 | lamu_GLEAN_10016673 | WRKYGQK | $C-X_4-C-X_{23}-HXH$ |
| I | MoWRKY8 | lamu_GLEAN_10019070 | WRKYGQK | $C-X_5-C-X_{23}-HXH$ |
| I | MoWRKY2 | lamu_GLEAN_10014815 | WRKYGQK | $C-X_5-C-X_{23}-HXH$ |
| I | MoWRKY3 | lamu_GLEAN_10006432 | WRKYGQK | $C-X_4-C-X_{23}-HXH$ |
| I | MoWRKY6 | lamu_GLEAN_10006277 | WRKYGQK | $C-X_5-C-X_{23}-HXH$ |
| I | MoWRKY4 | lamu_GLEAN_10010412 | WRKYGQK | $C-X_5-C-X_{23}-HXH$ |
| I | MoWRKY7 | lamu_GLEAN_10010176 | WRKYGQK | $C-X_5-C-X_{23}-HXH$ |
| I | MoWRKY5 | lamu_GLEAN_10005513 | WRKYGQK | $C-X_5-C-X_{23}-HXH$ |
| I | MoWRKY1 | lamu_GLEAN_10000767 | WRKYGQK | $C-X_5-C-X_{23}-HXH$ |
| I | MoWRKY10 | lamu_GLEAN_10018171 | WRKYGQK | $C-X_5-C-X_{23}-HXH$ |
| IIa | MoWRKY22 | lamu_GLEAN_10016899 | WRKYGQK | $C-X_5-C-X_{23}-HXH$ |
| IIa | MoWRKY23 | lamu_GLEAN_10005532 | WRKYGQK | $C-X_5-C-X_{23}-HXH$ |
| IIa | MoWRKY29 | lamu_GLEAN_10016902 | WRKYGQK | $C-X_5-C-X_{23}-HXH$ |
| IIb | MoWRKY26 | lamu_GLEAN_10015703 | WRKYGQK | $C-X_5-C-X_{23}-HXH$ |
| IIb | MoWRKY36 | lamu_GLEAN_10013925 | WRKYGQK | $C-X_5-C-X_{23}-HXH$ |
| IIb | MoWRKY30 | lamu_GLEAN_10010114 | WRKYGQK | $C-X_5-C-X_{23}-HXH$ |
| IIb | MoWRKY33 | lamu_GLEAN_10005737 | WRKYGQK | $C-X_5-C-X_{23}-HXH$ |
| IIb | MoWRKY38 | lamu_GLEAN_10016471 | WRKYGQK | $C-X_5-C-X_{23}-HXH$ |
| IIb | MoWRKY40 | lamu_GLEAN_10015347 | WRKYGQK | $C-X_5-C-X_{23}-HXH$ |
| IIb | MoWRKY39 | lamu_GLEAN_10018130 | WRKYGQK | $C-X_5-C-X_{23}-HXH$ |
| IIb | MoWRKY45 | lamu_GLEAN_10004479 | WRKYGQK | $C-X_5-C-X_{23}-HXH$ |
| IIc | MoWRKY17 | lamu_GLEAN_10015158 | WRKYGQK | $C-X_4-C-X_{23}-HXH$ |
| IIc | MoWRKY21 | lamu_GLEAN_10005936 | WRKYGQK | $C-X_4-C-X_{23}-HXH$ |
| IIc | MoWRKY18 | lamu_GLEAN_10014440 | WRKYGQK | $C-X_4-C-X_{23}-HXH$ |
| IIc | MoWRKY16 | lamu_GLEAN_10002123 | WRKYGQK | $C-X_4-C-X_{23}-HXH$ |
| IIc | MoWRKY50 | lamu_GLEAN_10005926 | – | $C-X_4-C-X_{23}-HXH$ |
| IIc | MoWRKY9 | lamu_GLEAN_10018985 | WRKYGQK | $C-X_4-C-X_{23}-HXH$ |
| IIc | MoWRKY14 | lamu_GLEAN_10013856 | WRKYGQK | $C-X_4-C-X_{23}-HXH$ |
| IIc | MoWRKY24 | lamu_GLEAN_10017233 | WRKYGKK | $C-X_4-C-X_{23}-HXH$ |
| IIc | MoWRKY13 | lamu_GLEAN_10016027 | WRKYGQK | $C-X_4-C-X_{23}-HXH$ |
| IIc | MoWRKY12 | lamu_GLEAN_10010840 | WRKYGQK | $C-X_4-C-X_{23}-HXH$ |
| IIc | MoWRKY44 | lamu_GLEAN_10009886 | WRKYGQK | $C-X_4-C-X_{23}-HXH$ |
| IIc | MoWRKY15 | lamu_GLEAN_10014128 | WRKYGQK | $C-X_4-C-X_{23}-HXH$ |
| IIc | MoWRKY51 | lamu_GLEAN_10003738 | WRKYGQK | $C-X_4-C-X_{23}-HXH$ |
| IIc | MoWRKY11 | lamu_GLEAN_10007141 | WRKYGQK | $C-X_4-C-X_{23}-HXH$ |
| IIc | MoWRKY19 | lamu_GLEAN_10017855 | WRKYGQK | $C-X_4-C-X_{23}-HXH$ |
| IId | MoWRKY31 | lamu_GLEAN_10007564 | WRKYGQK | $C-X_5-C-X_{23}-HXH$ |
| IId | MoWRKY28 | lamu_GLEAN_10011212 | WRKYGQK | $C-X_5-C-X_{23}-HXH$ |
| IId | MoWRKY27 | lamu_GLEAN_10016840 | WRKYGQK | $C-X_5-C-X_{23}-HXH$ |
| IId | MoWRKY25 | lamu_GLEAN_10013546 | WRKYGQK | $C-X_5-C-X_{23}-HXH$ |
| IId | MoWRKY20 | lamu_GLEAN_10005795 | WRKYGQK | $C-X_5-C-X_{23}-HXH$ |

**Table 1** (*continued*)

| Class | Gene name | Annotation ID | Conserved motify | Zinc finger |
|---|---|---|---|---|
| IIe | *MoWRKY47* | lamu_GLEAN_10007164 | WRKYGQK | $C-X_5-C-X_{23}-HXH$ |
| IIe | *MoWRKY35* | lamu_GLEAN_10001324 | WRKYGQK | $C-X_5-C-X_{23}-HXH$ |
| IIe | *MoWRKY37* | lamu_GLEAN_10016099 | WRKYGQK | $C-X_5-C-X_{23}-HXH$ |
| IIe | *MoWRKY42* | lamu_GLEAN_10013842 | WRKYGQK | $C-X_5-C-X_{23}-HXH$ |
| IIe | *MoWRKY46* | lamu_GLEAN_10012212 | WRKYGQK | $C-X_5-C-X_{23}-HXH$ |
| IIe | *MoWRKY32* | lamu_GLEAN_10009888 | WRKYGQK | $C-X_5-C-X_{23}-HXH$ |
| IIe | *MoWRKY48* | lamu_GLEAN_10014133 | WRKYGQK | $C-X_5-C-X_{23}-HXH$ |
| III | *MoWRKY52* | lamu_GLEAN_10005191 | WRKYGQK | $C-X_7-C-X_{23}-HXC$ |
| III | *MoWRKY41* | lamu_GLEAN_10009829 | WRKYGQK | $C-X_7-C-X_{23}-HXC$ |
| III | *MoWRKY34* | lamu_GLEAN_10014082 | WRKYGQK | $C-X_7-C-X_{23}-HXC$ |
| III | *MoWRKY49* | lamu_GLEAN_10012174 | WRKYGQK | $C-X_7-C-X_{23}-HXC$ |
| III | *MoWRKY54* | lamu_GLEAN_10006335 | WRKYGQK | $C-X_7-C-X_{23}-HXC$ |
| III | *MoWRKY53* | lamu_GLEAN_10005192 | WRKYGQK | $C-X_7-C-X_{23}-HXC$ |

## Structure analysis of *MoWRKY* genes

Intron/exon organization and numbers of introns are typical imprints of evolution within some gene families. In this study, we analysed the structure of *MoWRKY* genes to gain further insight into evolutionary events that had shaped them and found that all *MoWRKY* genes contain introns (Fig. 2A). The number of introns varies among genes, with the minimum, one intron, identified in five *MoWRKY* s (*MoWRKY50*, *MoWRKY44*, *MoWRKY51*, *MoWRKY19* and *MoWRKY11*) of subgroup IIc and the maximum, 10 introns, being present in *MoWRKY22*. Gene structure analysis revealed that genes with similar structures always clustered in the same class. For example, six members of group III all contained three exons and two introns. Similarly, five exons and four introns were present in *MoWRKY2*, *MoWRKY3*, *MoWRKY4*, *MoWRKY5* and *MoWRKY6*, which belonged to group I. However, the other five *MoWRKY* s in group I exhibited different gene structures.

## Motif composition analysis of MoWRKY proteins

The conserved motifs of WRKY proteins in drumstick were investigated using the MEME online software suite (http://meme-suite.org/) to better understand the similarity and diversity of motif compositions. Twenty distinct motifs were identified and a schematic overview of these motifs is provided in Fig. 3. For MoWRKY proteins, motif 1 was broadly distributed in all MoWRKY proteins, which was corresponded to WRKY domain. Motif 3 was only detected in the type I group. Motifs 5, 6, 7, 8, 9, 16, 17, 18 and 20 were only detected in the type II group; among them, motifs 5 and 6 were only detected in subgroup IIa and IIb, motifs 8 and 9 were only detected in subgroup IIb, motif 17 was only detected in subgroup IIc, and motif 16 was only detected in subgroup IId. Motifs 12 and 15 were only detected in the type III group. Generally, proteins with similar motif compositions were clustered in the same class indicating that members of the same class may have similar functions.

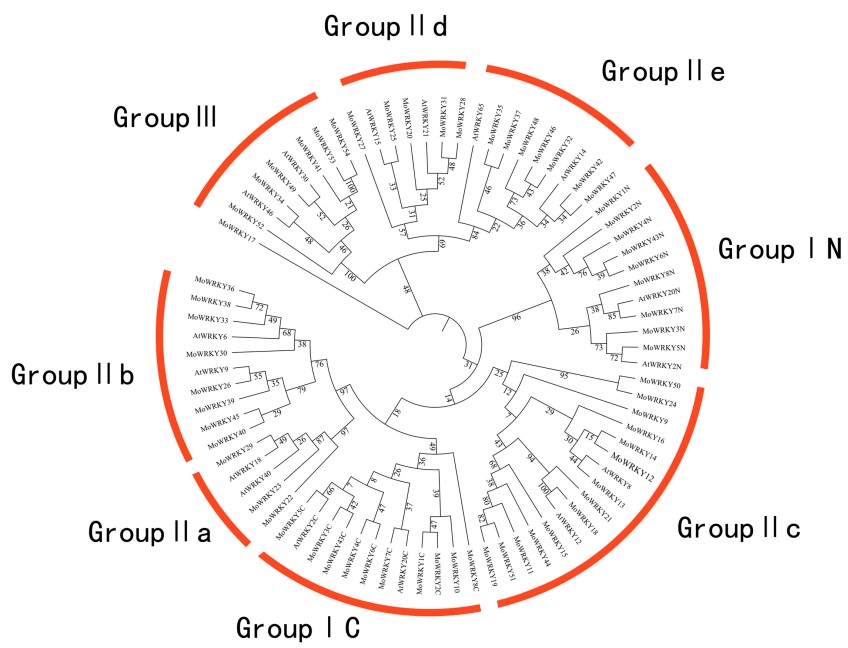

**Figure 1** **Phylogenetic tree of the WRKY conserved domain from drumstick and selected *Arabidopsis*.** The bootstrap test was performed with 1,000 replicates.

## Rapid expansion of group III WRKY genes in land plants

Group III WRKY genes have only been characterized in flowering plants, and a large number of duplications and diversifications in this group appear to have resulted from different selection challenges (*Dou et al., 2016*; *Huang et al., 2016*; *Kalde et al., 2003*). To explore the evolutionary relationships of group III WRKY genes across drumstick and other land plant species, we performed a multiple sequence alignment among the 81 group III WRKY proteins from drumstick and another seven species. A phylogenetic tree was constructed from the results of the alignment using the neighbor-joining method (Fig. 4). The marked difference in group III WRKY gene size among different species suggests that group III WRKY gene expansion occurred after the divergence of monocotyledons and dicotyledons. MoWRKY clearly shared more sequence similarity with VvWRKY and PaWRKY than with other WRKYs.

## The *cis-acting* elements analysis of *MoWRKYs*

For further understand the possible functions of *MoWRKY* genes, the cis-acting elements in all *MoWRKY* genes promoters were analyzed using PlantCARE software based the drumstick genome data. Various types of *cis-acting* elements were found and all *MoWRKY* genes contained several *cis-acting* elements in their promoter regions. The 10 most common elements were summarized in Table 2. These elements included three hormone responsive elements (ABRE, CGTCA motif and TGACG motif), an essential element for the anaerobic inductio (ARE), a drought stress responsive element (MBS), a heat stress responsive element (HSE) and four light responsive elements (Sp1, Box 4, G box and GT1 motif).

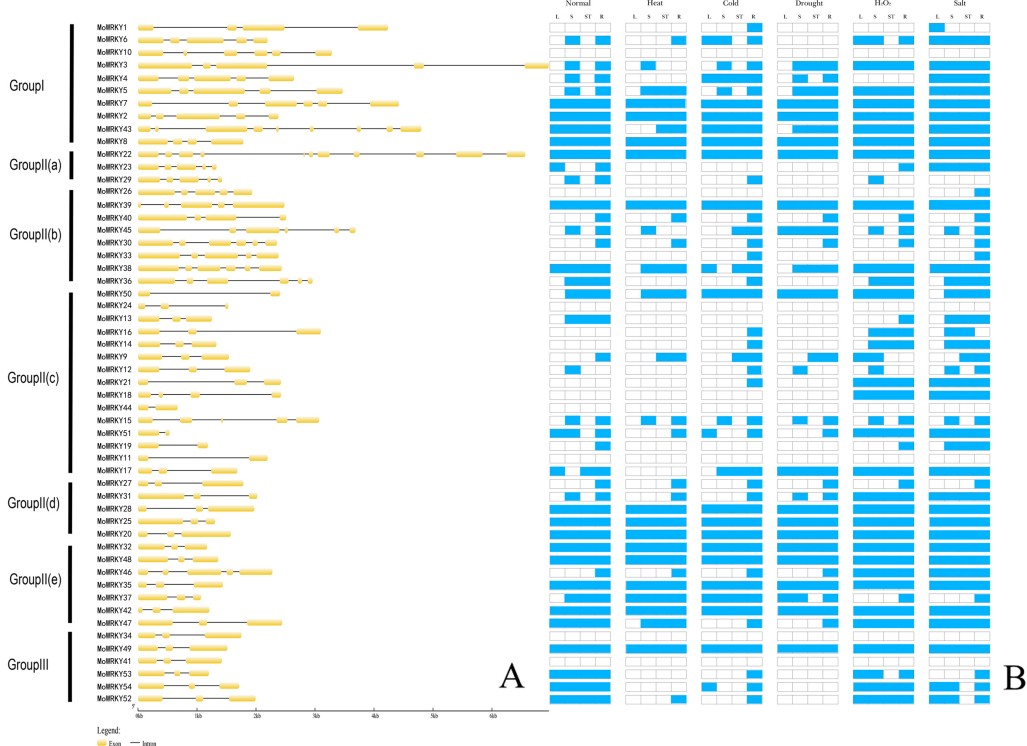

**Figure 2** **Exon-intron composition and expression patterns of *MoWRKY* genes.** (A) Exon-intron composition. (B) Expression patterns.

## Divergence in selective pressure between subgroups

The ratio ($\omega$) of the non-synonymous substitution rate ($Ka$) to the synonymous substitution rate ($K_S$) provides a sensitive measure of selective pressure acting on a protein-coding gene. Homologous genes with $\omega$ ratios of 1, <1, or >1 are usually assumed to be evolving under neutral evolution, purifying selection, or positive selection, respectively. To test for deviations in the substitution rates of *MoWRKY* genes, we calculated $\omega$ values across all pairwise comparisons within the 54 *WRKY* genes using the YN00 program in the PAML software package. The frequency distribution of $\omega$ values is shown in Fig. 5A. The results suggested that the *WRKY* gene family evolved mainly under strong purifying selection. However, there are several $\omega$ values greater than 1, such as those for the comparison between *MoWRKY8 and MoWRKY9 and that between MoWRKY8* and *MoWRKY10*, indicating that positive selection acted on these genes. Only 0.5% of the $\omega$ values approximated to 1, indicating that no selective pressure acted on these genes.

To test whether the rate of evolution among the subgroups of *WRKY* genes was identical, we calculated $\omega$ values across all pairwise comparisons within each of the subgroups; the results are shown in Fig. 5B. The average $\omega$ values of each subgroup were different. In order (highest first) they were: IIc, III, I, IIe, IId, IIb and IIa, indicating that different subgroups had evolved at different rates and that IIc had evolved the fastest.

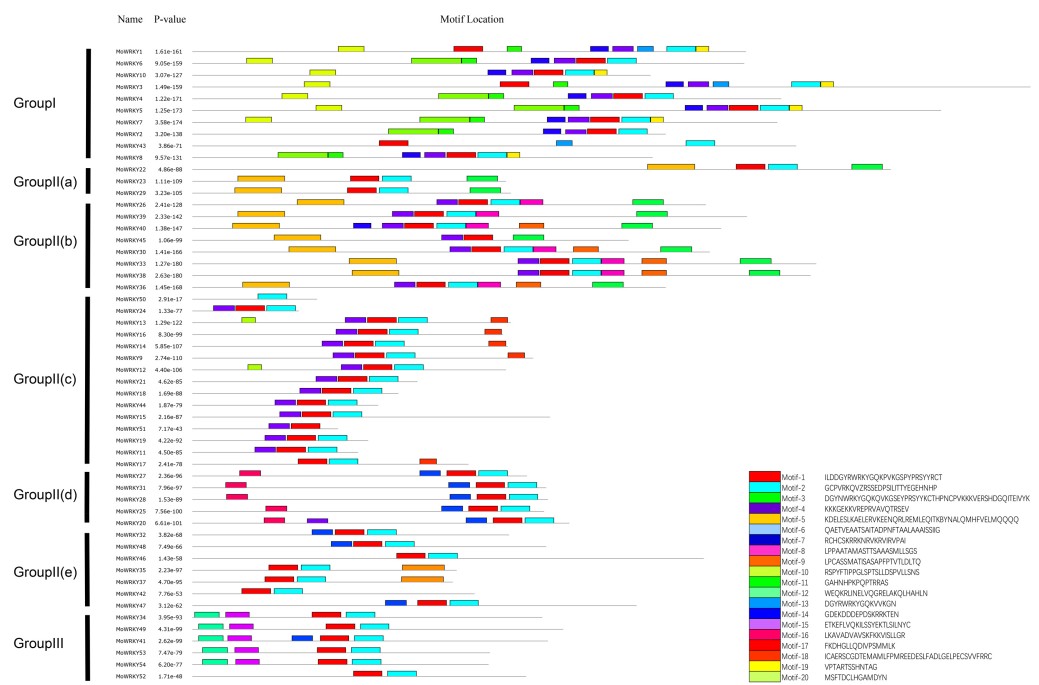

**Figure 3** **Distribution of conserved motifs in MoWRKYs.** Different colors represent different motifs.

## Expression patterns of WRKY genes in drumstick under normal growth conditions and abiotic stress conditions

To investigate the responses of *MoWRKY* genes to stresses, we examined the expression patterns of all 54 full-length *MoWRKY* s under normal growth conditions and under five abiotic stresses (heat, cold, drought, salt and oxidative) in different tissues (leaves, roots, stems, stem apex) using RT-PCR. As shown in Fig. 2B, among the 54 *MoWRKY* genes, 13 genes were expressed in all tissues under all growth conditions. In contrast, six genes, including *MoWRKY24*, the only gene with a variant WRKY domain (WRKYGKK), were not expressed in any tissue or in response to any of the treatments applied in this study. Thus, these six *WRKY* genes are expressed at undetectably low levels, or they are only induced in response to treatments and/or in tissues not examined in our study, or they are pseudogenes. The other 35 *WRKY* genes were expressed selectively in a specific tissue and/or in response to a specific treatment. Six of these genes were not expressed in any tissue under normal growth conditions but were expressed under stress conditions, suggesting that they play specific roles during stress conditions. At the same time, some genes, such as *MoWRKY46*, were only expressed in specific tissues under normal growth conditions but were expressed in all tissues under certain stress conditions, indicating that these genes may also play specific roles under stress conditions.

Nine *MoWRKY* genes from different subgroups were selected and their expression profiles were analyzed in root tissue under normal growth conditions and five abiotic stresses using qRT-PCR. As shown in Fig. 6, these selected *MoWRKY* genes were sensitive to abiotic stresses. All nine exhibited a high level of transcript accumulation under cold

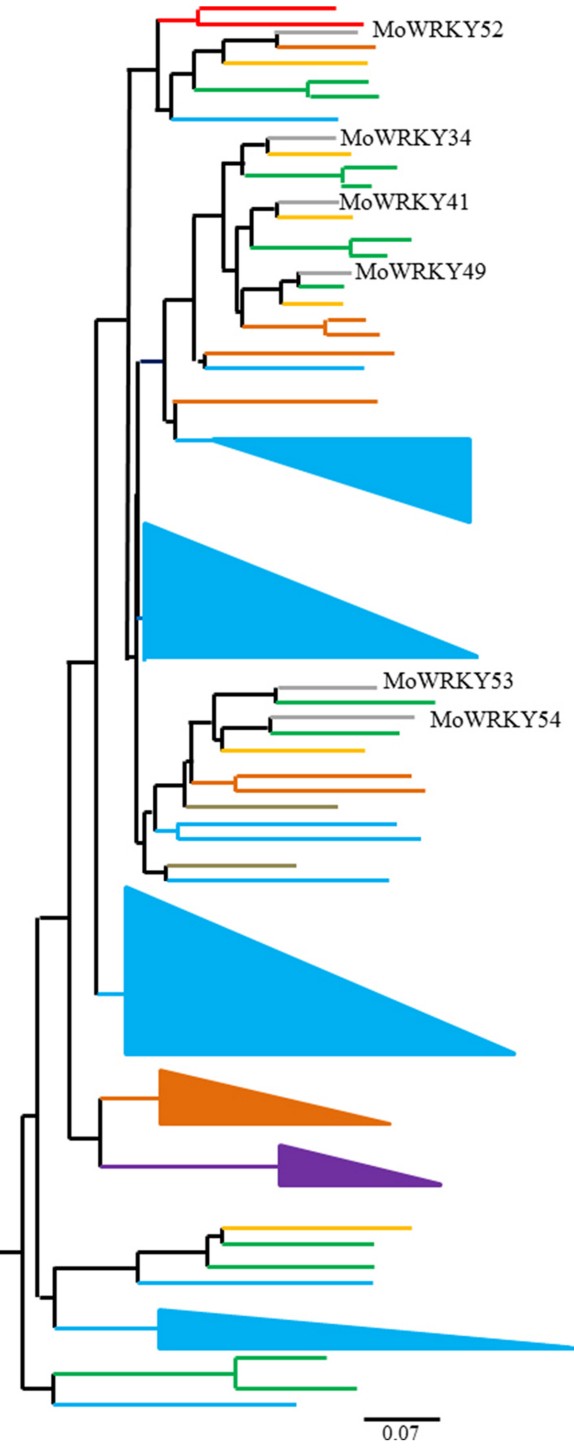

**Figure 4  Phylogenetic tree of 81 group III WRKY proteins from drumstick and other seven species.** *O. sativa* (blue triangles and lines), *P. euphratica* (green lines), *V. vinifera* (orange lines), *P. patens* (purple triangle), *A. thaliana* (brown triangle and lines), *S. moellendorfii* (grass green lines) and *P. abies* (red lines).

**Table 2  The predicted stress-responsive *cis-acting* elements in the promoters of *MoWRKYs*.**

| Cis-acting elements | Function | Genes |
|---|---|---|
| ABRE | Involved in ABA response | *MoWRKY1, 3, 4, 5, 6, 8, 9, 11, 13, 14, 15, 17, 19, 20, 21, 23, 24, 25, 26, 28, 29, 31, 33, 34, 37, 38, 40, 42, 43, 44, 46, 47, 48, 49, 51, 52, 53, 54* |
| ARE | Essential for the anaerobic induction | *MoWRKY 1, 2, 4, 5, 6, 7, 8, 9, 10, 12, 15, 16, 17, 18, 20, 21, 23, 24, 26, 27, 28, 30, 31, 32, 33, 34, 37, 38, 41, 42, 43, 45, 46, 47, 49, 51, 52, 53, 54* |
| MBS | Involved in drought inducibility | *MoWRKY 2, 4, 6, 7, 8, 10, 11, 12, 13, 14, 16, 17, 19, 20, 21, 22, 23, 24, 25, 26, 27, 28, 29, 30, 31, 32, 33, 34, 35, 36, 38, 39, 40, 42, 43, 44, 45, 47, 48, 49, 50, 53, 54* |
| HSE | Involved in heat stress response | *MoWRKY 3, 4, 5, 9, 12, 14, 16, 19, 20, 21, 22, 23, 26, 27, 28, 29, 30, 32, 33, 35, 36, 38, 39, 40, 41, 43, 45, 46, 47, 49, 50, 51, 53* |
| Sp1 | Light responsive element | *MoWRKY 4, 5, 6, 9, 10, 11, 13, 14, 18, 19, 20, 21, 23, 24, 27, 28, 31, 35, 37, 38, 39, 41, 42, 44, 46, 47, 48, 50, 52, 53, 54* |
| G-box | ABA, light, UV and hurt responsive element | *MoWRKY 1, 2, 3, 4, 5, 6, 7, 8, 9, 10, 11, 12, 13, 14, 15, 16, 17, 19, 20, 21, 23, 24, 25, 26, 28, 29, 31, 32, 33, 34, 35, 36, 37, 38, 39, 40, 42, 43, 44, 45, 46, 47, 48, 49, 51, 53, 54* |
| Box 4 | Part of a conserved DNA module involved in light response | *MoWRKY 1, 2, 3, 4, 6, 8, 9, 10, 11, 12, 13, 14, 17, 18, 19, 20, 21, 22, 23, 25, 26, 27, 28, 29, 30, 31, 32, 33, 34, 35, 36, 37, 38, 39, 40, 41, 42, 43, 44, 45, 46, 48, 49, 50, 51, 52, 53* |
| CGTCA motif | Involved in MeJA response | *MoWRKY 1, 3, 4, 6, 7, 8, 9, 11, 12, 15, 16, 17, 18, 19, 21, 22, 24, 25, 27, 28, 29, 30, 33, 34, 35, 37, 38, 39, 40, 41, 43, 44, 46, 48, 49, 50, 51, 52, 53, 54* |
| TGACG motif | Involved in MeJA response | *MoWRKY 1, 3, 4, 6, 7, 8, 11, 12, 15, 16, 17, 18, 19, 21, 22, 24, 25, 27, 28, 29, 30, 33, 34, 35, 37, 38, 39, 40, 41, 43, 44, 46, 48, 49, 50, 51, 52, 53, 54* |
| GT1 motif | Light responsive element | *MoWRKY 1, 6, 7, 9, 11, 12, 13, 15, 16, 18, 19, 20, 22, 23, 24, 26, 27, 29, 34, 35, 36, 37, 39, 40, 43, 44, 45, 47, 48, 49, 50, 54* |

stress, especially *MoWRKY30* (GO: 0006950), followed by *MoWRKY54* (GO: 0006950 and GO: 0080134). Interestingly, the genes that were most strongly up-regulated under cold treatment were always up-regulated in response to heat and salt treatments. In drought stress, *MoWRKY22*, which had the most introns and *MoWRKY3*, which was the longest in *MoWRKY* gene family were found to be slightly upregulated, whereas weak expression were found for the other seven genes. The expression levels of almost all the nine *MoWRKY* s were decreased under oxidative stress. *MoWRKY49*, *MoWRKY53* and *MoWRKY54*, which all belonged to group III, have similar gene structures and the same motifs. But the expression levels of the three genes under abiotic stresses were slightly different. They were evidently upregulated in cold and salt to different degrees; *MoWRKY53* (GO: 0006950 and GO: 0080134) and *MoWRKY54* (GO: 0006950 and GO: 0080134) were also responsive to heat. Overall, the expression patterns of *MoWRKY* s under various conditions suggest that different *MoWRKY* genes may be involved in different signaling and stress responses, and that an individual *MoWRKY* gene can also participate in multiple signaling and stress process.

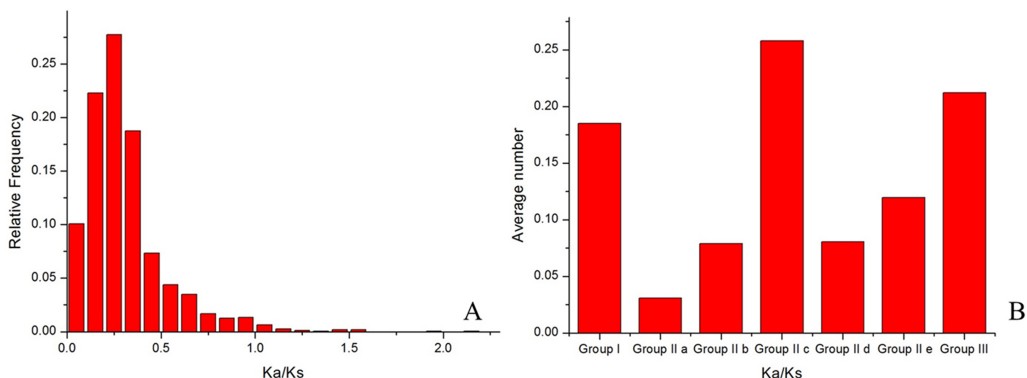

**Figure 5  Frequency distribution and average values of Ka/Ks ratios.** (A) Frequency distribution between any two drumstick *WRKY* genes. (B) Average values of Ka/Ks across sub-groups of drumstick *WRKY*s.

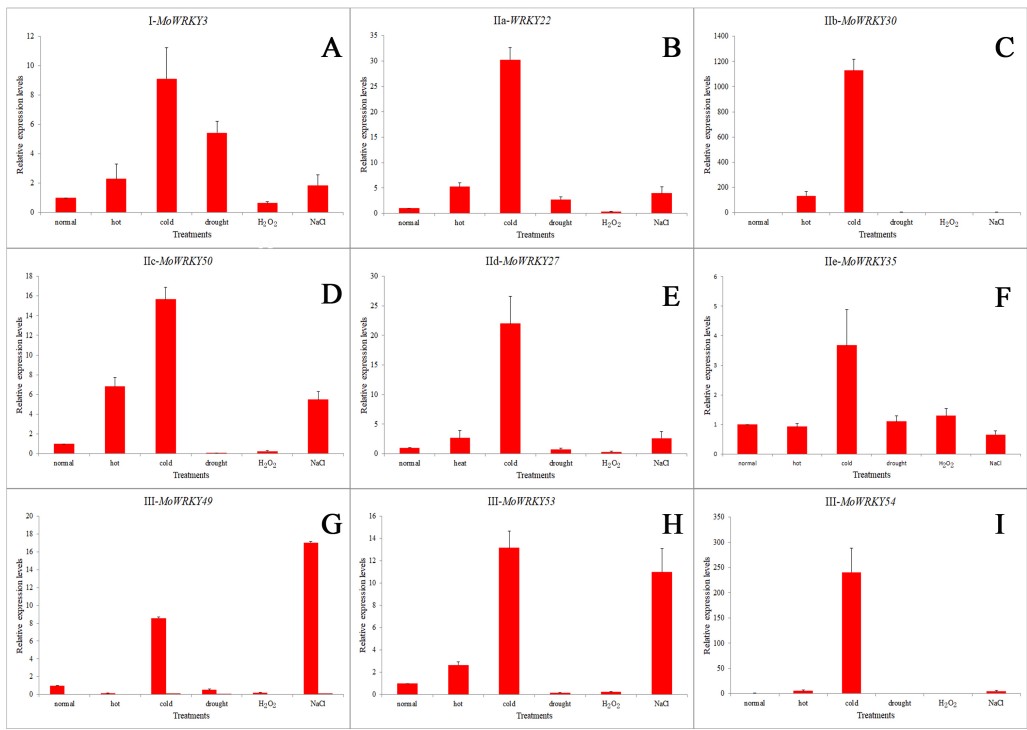

**Figure 6  Expression profiles for nine selected *MoWRKY* genes in root under different stresses.** (A) *I-MoWRKY3*; (B) *IIa-MoWRKY22*; (C) *IIb-MoWRKY30*; (D) *IIe-MoWRKY50*; (E) *IId-MoWRKY27*; (F) *IIe-MoWRKY35*; (G) *III-MoWRKY49*; (H) *III-MoWKRY53*; (I) *III-MoWKRY54*.

## DISCUSSION

WRKY transcription factors were first identified over 20 years ago (*Ishiguro & Nakamura, 1994*) and it has been suggested that they play important roles in stress responses and at many stages of plant growth and development (*Phukan, Jeena & Shukla, 2016*; *Tripathi,*

*Rabara & Rushton, 2014*). Genes encoding WRKY proteins belong to a large family, with 72 members in *Arabidopsis thaliana* (*Wang et al., 2011*), 100 in *Oryza sativa* (*Ross, Liu & Shen, 2007*) and 104 members in the *Populus trichocarpa* genome (*He et al., 2012*). A previous study showed that *Populus trichocarpa* (*He et al., 2012*) and *Daucus carota* (*Li et al., 2016*) WRKYs could be divided into three groups. In the present study, when a phylogenetic tree of WRKYs from drumstick and *Arabidopsis* was constructed, we found that the 54 WRKYs from drumstick fell into three distinct groups. This result was consistent with the WRKY domain and zinc finger type classification of these WRKYs. When the subgroups of WRKY genes were compared among *Arabidopsis*, rice and poplar, we found that the number of each subgroup in group II was similar indicating that all members of these subgroups have probably been identified. However, the number of *MoWRKY* s in group III is less than the numbers in *Arabidopsis* and rice which are older species, implying that *WRKY* genes of this group in drumstick either had been lost during the course of evolution or were underrepresented in our analysis.

The WRKY conserved domain is the most important functional and evolutionary unit of WRKY transcription factors. Although the WRKYGQK amino acid residues in the WRKY domain are highly conserved, there are variants. Six sequence variations (WRKYGHK, WRKYGQN, WRKYGKK, WRKCGQK, WRKYGQT, WRKYGMK) were found in *CaWRKY* genes. Six heptapeptide variants, namely WRKYGKK, WRKYGEK, WRKYGKR, WRKYEDK, WKKYGQK, WHQYGLK, were found in soybean (*Song et al., 2016*). In our study, only one variant (WRKYGKK) was found, and only in *MoWRKY24* which belongs to subgroup IIc. WRKYGKK is the most common variant in many species. In the tobacco WRKY protein family, the WRKYGKK domain could bind specifically to a WK-box, which was significantly different from the W-box (*Verk et al., 2008*). In our study, we could not detect the expression of *MoWRKY24* in any tissues or under any stress conditions. The reason may be that the expression level of *MoWRKY24* was too low to be detected, or that this gene is only expressed under special conditions, or that it has become a pseudogene. This apparent lack of expression needs to be investigated further.

The structures of the *MoWRKY* genes showed group-specific exon-intron patterns, as is also the case in carrot (*Li et al., 2016*) and cassava (*Wei et al., 2016*). Exon-intron structural diversity plays an important part in the evolution of gene families (*Wei et al., 2016*). The number of introns in *MoWRKY* genes varied from 1 to 10. However, in poplar (*He et al., 2012*) and cassava (*Wei et al., 2016*), the number of introns varied from, respectively, 0 to 6 and 1 to 5. The results indicated that *MoWRKY* s have more gene structure diversity than the poplar and cassava *WRKY* genes. In our study, the length of the *MoWRKY3* gene in group I was greater than those of any other genes. While neither the number nor the length of exons in this gene was unusually high, there were more introns. Combined motif compositions, we can find the variety and average length of motifs identified in MoWRKY3 were not especially large, indicating that their functions were probably not influenced by the presence of the numerous introns. According to a previous report, the rate of intron loss is faster than the rate of intron gain after segmental duplication (*Nuruzzaman et al., 2010*) and intron loss can result from intron turnover or reverse transcription of the mature mRNA followed by homologous recombination with intron-containing alleles

(*He et al., 2012*). In drumstick, members of group III all contained two introns; the average number of introns in the other groups was more than that in this group. Consequently, it can be inferred that group III developed later than other groups. The structure and motif compositions of group III members were very similar, indicating that these genes expanded not by merging, transfer or loss but in other ways.

WRKY proteins usually functioned as transcriptional regulators by binding to W-box to regulate defense-related genes. In our study, we found that nearly half *MoWRKY* genes also contained W-box element in their promoter regions. The same findings were identified in carrot (*Li et al., 2016*) and soybean (*Song et al., 2016*), suggesting that these *MoWRKY* genes are auto-regulated by themselves or cross-regulated. Accumulating evidence suggests that WRKY transcription factors are involved in many plant processes including development and responses to biotic and abiotic stresses and that may due to the upstream genes specificity bind the corresponding cis element to regulate the expression of *WRKY* genes. In carrot, fourteen selected *DcWRKY* genes responded to whitefly and aphid infections and twelve *DcWRKY* genes were upregulated or downregulated under heat and/or cold treatments (*Li et al., 2016*). At least 31 *PeWRKY* genes in moso bamboo (*Li et al., 2017*) and 21 *CaWRKY* genes in pepper (*Diao et al., 2016*) were differentially expressed under abiotic stresses. Similarly, 55 *VvWRKY* genes in grape (*Zhang & Feng, 2014*) differentially responded to at least one abiotic stress treatment. In our study, the results of expression pattern analysis demonstrated that most *MoWRKY* genes had different expression levels when the seedlings were exposed to different stresses despite highly homologous amino acid sequences and conserved domain structures. *WRKY* genes within the same group may act as redundant and substitute members in regulating functions. The very large expression differences suggested that the products of these genes have different physiological functions, facilitating adaptation to complex challenges. Further structural analyses and investigations into the expression patterns of the *MoWRKY* gene family would facilitate a more comprehensive understanding of the specific functions of individual WRKY genes. The current investigation highlights a number of *MoWRKY* genes that may be involved in stress defenses, and lays a solid foundation for the selection of candidate genes for further studies.

## CONCLUSION

The publication of drumstick genome sequences provides an opportunity for genome wide identification and characterization of WRKY TFs. Bioinformatics tools have been made in the present study to identify the putative members of WRKY genes of drumstick and subject it to characterization for gene structures, motif analysis, conserved motifs and phylogenetic tree construction. The multiple members of *WRKY* genes in plants reflect the redundancy and differentiated functions of these proteins which need to be explored by expression profiling. The expression profiling under different abiotic stress conditions revealed several potential MoWRKYs showing higher expression level under drought, salt, cold and heat stresses.

## ACKNOWLEDGEMENTS

The authors would like to thank professor Jun Sheng and Yang Tian at Yunnan Agricultural University for their kind sharing of the drumstick genome sequences data. The authors also would like to thank the reviewers for their careful reviewing and helpful comments on the manuscript.

### Funding

This work was funded by the Forestry Technology Innovation Program, the Department of Forestry of Guangdong Province (2015KJCX009; 2017KJCX029), the Guangzhou Science Technology and Innovation Commission (201707010462), and the Characteristic Innovation Program (Natural Science), the Department of Education of Guangdong Province (2018KTSCX018). The funders had no role in study design, data collection and analysis, decision to publish, or preparation of the manuscript.

### Grant Disclosures

The following grant information was disclosed by the authors:
Forestry Technology Innovation Program.
Department of Forestry of Guangdong Province: 2015KJCX009, 2017KJCX029.
Guangzhou Science Technology and Innovation Commission: 201707010462.
Characteristic Innovation Program (Natural Science), the Department of Education of Guangdong Province: 2018KTSCX018.

### Competing Interests

The authors declare there are no competing interests.

### Author Contributions

- Junjie Zhang conceived and designed the experiments, performed the experiments, prepared figures and/or tables, authored or reviewed drafts of the paper, approved the final draft.
- Endian Yang performed the experiments, prepared figures and/or tables.
- Qian He performed the experiments, approved the final draft.
- Mengfei Lin analyzed the data, prepared figures and/or tables.
- Wei Zhou analyzed the data, contributed reagents/materials/analysis tools.
- Ruiqi Pian analyzed the data.
- Xiaoyang Chen conceived and designed the experiments, contributed reagents/materials/analysis tools, authored or reviewed drafts of the paper, approved the final draft.

### Data Availability

The raw data is available as Supplemental File. The raw data shows CDS sequences and amino acid sequences of MoWRKY.

All the data was downloaded from common databases. The gene and proteome sequences and General Feature Format (GFF) file of WRKY gene family for Arabidopsis were downloaded from TAIR (https://www.arabidopsis.org/servlets/Search?type=general&search_action=detail&method=1&show_obsolete=F&name=WRKY&sub_type=gene&SEARCH_EXACT=4&SEARCH_CONTAINS=1).

A WRKY-domain Hidden Markov Model (HMM) Profile was downloaded from Pfam (http://pfam.xfam.org/).

## Supplemental Information

Supplemental information for this article can be found online at http://dx.doi.org/10.7717/peerj.7063#supplemental-information.

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
