# Peer review of "Genome-wide analysis of the WRKY gene family in drumstick (Moringa oleifera Lam.)"

_PeerJ, doi:10.7717/peerj.7063_

## Round 0.1 · original submission · Major Revisions

· Academic Editor

Major Revisions

Both the reviewers have raised several questions. All the questions has to be addressed by the authors before further consideration.

Reviewer 1 ·

Basic reporting

No comment

Experimental design

In-depth bioinformatic analysis executed by the authors. However, lack of experiment to address functional role of gene of interest.

Manuscript needs to further discuss and enrich the functional role of MoWRKY genes and specific motifs in relevance to its genome distribution.

Validity of the findings

To enhance impact and novelty of their study, the current version of manuscript needs minor revision.

Additional comments

In the current manuscript by Zhang et.al. “Genome wide analysis of WRK gene family in Drumstick (Moringa oleifera Lam.)”, authors discuss the role of MoWRKY genes importance using bioinformatic tool complemented along with differential expression profiling under different stress conditions.
WRKY gene family is assumed to have different function in diverse plant function and process. However, authors do not discuss the precise and specific role of WRK gene family in Drumstick. To consider the manuscript for publication quality authors should emphasize and clearly discuss the role and function of such gene in Moringa oleifera. Having redundant expression and functional activity, it is not clear how MoWRKY gene is regulated under different stress condition. What is the functional relevance of such differential expression and having diverse functional role? It should be discussed in detail in context to their structural analysis.

Reviewer 2 ·

Basic reporting

This sections seems fine

Experimental design

Need to work on Experimental design ---especially for expression of the mentioned genes

Validity of the findings

WRKY proteins are DNA binding proteins, characterized by presence of WRKY motif and ZnF. WRKY proteins constitute one of the largest family of transcription factors and have been attributed to many functions in plant such as stress response and signal transduction. The present manuscript describes the identification and classification of WRKY gene family in drumstick. Authors have also reported the stress effects on the expression of these genes. The abstract and introduction sections are nicely written. However, authors need to work on method and figures. I have following major concerns:
1) Why authors haven’t shown the clustalW data? It would be good to see how they are conserved.
2) Variant WRKYGKK seems to be particularly interesting as it is not expressed in any tissue in any of the studied conditions. It would be interesting to know how a single residue (Q to K) mutation leads to complete abolishment of the function. Does this variant evolve to behave as paralogue?
3) In Figure 6, Y-axis is unlabeled. What is the proxy for expression measurement? If the numbers on Y-axis are normalized, it should be clearly mentioned in legend and methods.
4) Line 252 “Nine MoWRKY genes from different subgroups were selected…”. What was the basis for selection of these genes? Authors should clearly mention that and should be justified (supplementary?).
5) How were the error bars calculated in Figure 6? This should be clearly stated in method/legend.
6) What is the error range in Figure 5? What do Ka and Ks stand for? It would be difficult for readers to follow these.
7) Resolution of Figure 2&3 is very poor. The labels are barely clear.
8) Method section should be more elaborated.

---

## Round 0.2 · Minor Revisions

· Academic Editor

Minor Revisions

Authors have incorporated all the suggested changes. The manuscript is close to being for accepted for publication, however Gerard Lazo, the Section Editor for this article has made the additional request/statement:

“The manuscript reads well, but is lacking in updating the gene family characterization by using an ontology annotation. Journal manuscripts are often scanned by text-mining software that locates and extracts core data elements, like gene function. Adding standard ontology terms, such as the Gene Ontology (GO, geneontology.org) or others from the OBO foundry (obofoundry.org) can enhance the recognition of the contribution and description. This will also make human curation of literature easier and more accurate.

GO terms are a strong attribute tied in with the Arabidopsis and many other communities, and as these resources were cited it would be a complementing service to do the same, and would heighten the value of the manuscript. It is the use of such terms which have moved the technology further in recent years. As different abiotic stresses were addressed and the fine detail of the gene structure was also addressed, adding appropriate GO terms would be very helpful to the readership of this well-described transcription factor family."

Therefore, this is a Minor Revision decision allowing you to to explore adding ontology terms to the document. As this is a characterization of a well studied transcription factor family, the adding of ontology terms will benefit the research community and improve the impact factor of the manuscript.

Reviewer 1 ·

Basic reporting

Revised manuscript is address clearly.

Experimental design

Experimental design is executed well to address the objective.

Validity of the findings

Well defined.

Reviewer 2 ·

Basic reporting

The authors have addressed majority of my concerns and hence can be accepted.

Experimental design

The authors have addressed majority of my concerns and hence can be accepted.

Validity of the findings

The authors have addressed majority of my concerns and hence can be accepted.

Additional comments

The authors have addressed majority of my concerns and hence can be accepted.

---

## Round 0.3 · accepted · Accept

· Academic Editor

Accept

Gene Ontology (GO) terms of WRKY gene family has been added. Manuscript is now ready for publication.